# A simulation study demonstrating the importance of large-scale trailing vortices in wake steering

Paul Fleming[1], Jennifer Annoni[1], Matthew Churchfield[1], Luis Martinez[1], Kenny Gruchalla[1], Michael Lawson[1], and Patrick Moriarty[1]

[1]National Wind Technology Center, National Renewable Energy Laboratory, Golden, CO, 80401, USA

*Correspondence to:* Paul Fleming (paul.fleming@nrel.gov)

**Abstract.** In this paper, we investigate the role of flow structures generated in wind farm control through yaw misalignment. A pair of counter-rotating vortices is shown to be important in deforming the shape of the wake and in explaining the asymmetry of wake steering in oppositely signed yaw angles. We also demonstrate that vortices generated by an upstream turbine that is performing wake steering can deflect wakes of downstream turbines, even if they are themselves aligned.

We motivate the development of improvements to control-oriented engineering models of wind farm control, to include the effects of these large-scale flow structures. Such a new model would improve the predictability of control-oriented models. Further, we demonstrate that the vortex structures created in wake steering can lead to greater impact on power generation than currently modeled in control-oriented models. We propose that wind farm controllers, can be made more effective if designed to take advantage of these effects.

## Copyright Statement

## 1 Introduction

Wake steering is a wind farm control concept where the upstream turbines are intentionally misaligned to deflect its wake away from a downstream turbine (Wagenaar et al. (2012).) For certain arrangements of turbines, it can be shown that the power of the downstream turbine is increased by more than is lost by misaligning the upstream, yielding a net increase in power.

Early research in this field used wind tunnel experiments to demonstrate the possibility of wake steering. Experiments presented in Dahlberg and Medici (2003) showed that for example, wake steering implemented in a two turbine row could

yield a total relative power gain of 10%. In Medici and Alfredsson (2006), the wake of a model wind turbine in a tunnel was measured. When the turbine was yawed, the wake was deflected, and additionally, vortex shedding was observed which was similar in behavior to what would be expected from solid discs. In Wagenaar et al. (2012), wake steering is studied at a scaled wind farm.

Later work in wake steering made assessments of the technique using computational fluid dynamic (CFD) simulations of small arrays of wind turbines. For example, Fleming et al. (2015) examined wake-steering performance for an offshore array of two turbines separated by 7 diameters, and observed an increase in total power.

In order to design controllers to implement wake-steering control strategies, it is necessary to build engineering models that contain the relevant physics but are described in a computationally efficient way such that the model can be used in large-scale

optimizations or used as an internal model within a real-time controller. A first model of wake steering, based on CFD analysis, was provided by Jiménez et al. (2010). The model predicts wake deflection as a function of yaw angle and thrust coefficient.

In Gebraad et al. (2016), an engineering wake model, known as FLORIS (FLow Redirection and Induction in Steady-State) was introduced. The model combined the wake-steering model of Jiménez et al. (2010) with the wake recovery model of Jensen (1984). In addition, this model separates the wake into zones that recover at different rates. The model was shown to predict

the behavior of wake steering for a given set of CFD simulations (focused largely on two-turbine, and fully waked scenarios) and, given its execution speed, can be used to design controllers as well as look at coupled wind farm layout and controls optimizations (c.f. Gebraad et al. (2017)). In this model, the wake appears as a deficit of energy that flows downstream and is characterized as (i) an initial deficit determined by the turbine thrust, (ii) a rate of recovery determined by topology, i.e. FLORIS inherits wake-recovery parameters from the Jensen model whose values are set by rules of thumb for onshore/offshore, and (iii)

a wake deflection caused by wake-steering. This version of FLORIS was used to design a strategy for wake steering that was tested at a commercial offshore wind farm in Fleming et al. (2017b) with an improved wind farm power performance under waked conditions.

More recently, the FLORIS model has been improved by incorporating the theoretical models of wake behavior and steering presented in Bastankhah and Porté-Agel (2014, 2016); Niayifar and Porté-Agel (2015). One of the improvements was the

inclusion of turbulence to describe the wake recovery. Turbulence in turbine wakes is generated by ambient wind conditions as well as turbine operating conditions. These critical changes greatly improve the general applicability of models such as FLORIS. For example, wake expansion and the recovery are now dependent on ambient and induced turbulence intensity, which can resolve the modeling discrepancies observed in Annoni et al. (2016).

To date, engineering models are beginning to include modeling of the main atmospheric components (turbulence intensity,

veer, and shear) needed to accurately predict and control wakes via wake steering. The basic conception of wake steering remains that a rotor operating in misaligned conditions creates a force perpendicular to the flow that generates the deflection in wake direction and decreases the thrust and overall velocity deficit in the wake. Further, based on Bastankhah and Porté-Agel (2014, 2016); Niayifar and Porté-Agel (2015), the change in thrust also impacts recovery rate. Wind tunnel tests in Schottler et al. (2017) and Bartl and Sætran (2017) show good agreement overall with these latest models.

However, recent research has underlined that in addition to changing the deficit and location of a wake, yaw misalignment changes other properties in the flow, which can be observed as impacting the shape of the wake. In Vollmer et al. (2016), large-eddy simulations (LES) are used to study the behavior of wake steering under varying atmospheric conditions. The authors note that the shape of the wake under yawed conditions is curled rather than circular. This curvature is shown to impact the estimate of the wake center. Further, the authors explain the change in shape by showing that the cross-flow wind of an aligned turbine is largely due to counter-rotating vortices that appear in the flow behind a yawed turbine and generate this distortion. This is consistent with the results of Medici and Alfredsson (2006).

Howland et al. (2016) studies the curled wake phenomenon experimentally using a porous actuator disk and with LES using an actuator disk and an actuator line model. The curled wake is observed in experiments and simulations. The mechanism behind the curled wake is again explained as a pair of counter rotating vortices which are shed from the top and bottom of the rotor due to yaw misalignment. This mechanism was confirmed by Bastankhah and Porté-Agel (2016).

In Fleming et al. (2017a), a rear-facing nacelle-mounted lidar is used to scan the wake of a turbine in aligned and yawed conditions at five locations downstream ranging from 1 diameter to approximately 3 diameters. The deflection predicted by earlier simulation models is clearly observed as well as the curling of the wake shape.

The current state of control-oriented models supposes that the change in wake "location" and strength can be well modeled as a deflection of a wake generated with a lower thrust. In other words, the effect of the counter-rotating vortices on the flow is assumed to be captured in the engineering models that modify only the deflection, deficit and recovery of the wake. A curled wake of some deflection amount might be well-enough described by a circular wake of larger deflection.

In this paper, a CFD-based analysis is used to examine how the consideration of the counter-rotating vortices can impact wind farm control analysis and design. This paper undertakes an investigation of the impact of these vortices on yaw-based wake control. The contributions of this paper are first a demonstration that a deflection-only control-oriented model of wind farm control can not reconcile all observed effects, even, to some extent, for a single turbine wake case. A second contribution is the demonstration that the influence of the vortices is especially critical when arrays of multiple turbines are considered. A steered wake of an upstream turbine is shown to deflect the wake of an aligned turbine downstream, and combinations of steered turbines are shown to involve merging of generated cross flows. The discussion section considers how wind farm control, based on the generation of specific large-scale structures, and not on geometrical deflection, could be different and more effective than current methods. The future work recommendations conclude that incorporating the shed vortices into engineering models used to design wind farm controllers can improve wind farm control performance and should be undertaken.

## 2   Models

This paper focuses on two specific wind farm models. First, FLORIS, is a low-fidelity, control-oriented tool for wind farm control, which includes several possible wake models developed by NREL and TU Delft Gebraad et al. (2016). FLORIS is python-based, open source, and available for download on github (https://github.com/WISDEM/FLORIS). The overall approach of FLORIS is to provide an engineering model of wakes that predicts the important average behaviors of wakes in

a computationally efficient way such that it can be used to derive control strategies through optimization or function as an internal controller model. A recent report compares predictions of the latest FLORIS model to lidar data from a utility-scale wind turbine operating in yaw and provides good agreement (Annoni et al. (2018)).

In this work, the wake model used in this paper assumes a Gaussian wake that is derived from self-similar turbulence theory ( Bastankhah and Porté-Agel (2014, 2016); Niayifar and Porté-Agel (2015)). The wake expands linearly and the parameters of this Gaussian wake are a function of ambient turbulence intensity and turbine operation. Overlapping wakes are combined using a sum-of-squares approach that has been used previously in literature (Katić et al. (1986)). It is important to note that there are alternative methods to combine wakes as wake superposition is an ongoing research topic (Machefaux and Mann (2015), Trabucchi et al. (2017)).

Wake deflection is also included in this model based on the yaw misalignment of a turbine. It is modeled using a budget analysis of the Reynolds Averaged Navier-Stokes equations. For further details, the reader is referred to (Bastankhah and Porté-Agel (2016)).

Second, the Simulator fOr Wind Farm Applications (SOWFA), is a high-fidelity framework used to perform large-eddy simulations (LES) of wind farm flows, developed by NREL (Churchfield et al. (2012)). In LES, the filtered Navier-Stokes equations are solved numerically, providing a time-evolving three-dimensional flow field in the wind farm. Within SOWFA, wind turbines are modeled using an actuator disk/line model with torque, blade pitch, and yaw controllers. SOWFA is based on the OpenFoam libraries, and has been used in past research of wind farm controls for design and analysis (Churchfield et al. (2012)).

In past research, the SOWFA and FLORIS have been used together. In a typical workflow, SOWFA simulations of a small farm are run to provide tuning inputs to FLORIS, which is then used to predict optimal controllers that are then tested in SOWFA. Successful iterations of this process yield controllers that can be used in field testing (see for example (Gebraad et al. (2016); Fleming et al. (2017b))

One constraint of this approach is that, given the computational requirements of running SOWFA, tuning cases are typically limited to approximately 10-20 runs, and these are focused on high-loss scenarios such as one turbine directly waking another. For example, simulations of two turbines aligned in the flow, with and without yaw, for different distances downstream are a typical focus.

An example of the approach used in this paper is now described. In one scenario, a SOWFA simulation including one turbine modeled as an actuator disk with rotation is simulated (see Fig 1). Fig 1 illustrates a turbine in positive yaw in the convention used of positive being a counter-clockwise rotation when viewed from above. Note that for all figures, horizontal planes (such as Fig 1) are viewed from above, while cross-stream planes (such as Fig. 3) are viewed from upstream. The simulation is run for 2,000 seconds, and the flow is averaged over the last 1600 seconds to allow for transients to dissipate. The averaged flow is provided directly via OpenFOAM output functions. From the averaged flow, a power is computed, which would have been produced by an additional hypothetical turbine at some point downstream. This way, rather than results from a handful of turbine positions, a continuum of turbine locations can be considered. This approach was adapted from power calculations used in Vollmer et al. (2016) This power is computed by averaging the cubed wind speed over a hypothetical rotor disk from

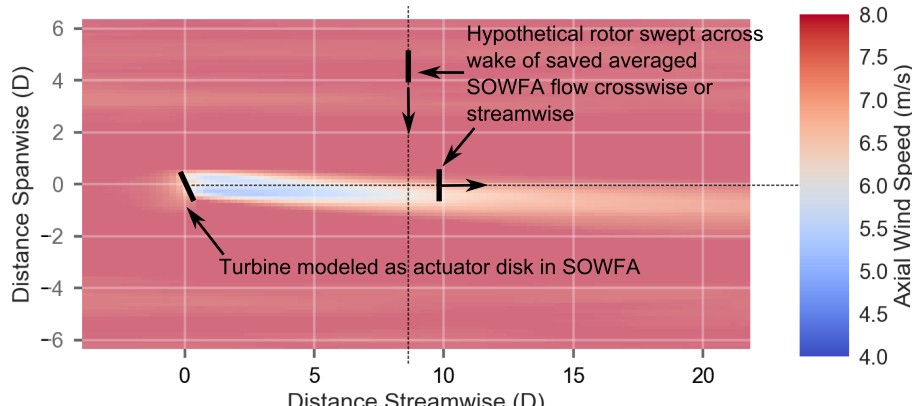

**Figure 1.** Diagram of the method used in the paper. In this example, showing the flow as viewed from above, a single turbine is included in a SOWFA simulation. A period of 1600 seconds of the flow is averaged and then the hypothetical turbine rotor is scanned across this averaged flow to compute the power that would have been produced by a turbine at a given location.

the averaged flow. The cubed average is then converted to power through

$$P = 0.5\rho A C_p U^3 \tag{1}$$

where $\rho$ is the density, $A$ is the rotor area, $C_p$ is the coefficient of power (which is derived from a look-up table based on wind speed precomputed from the aero-elastic turbine simulator FASTJonkman and Buhl Jr. (2005)), and $U$ is the rotor averaged
wind speed. The hypothetical turbine can be swept across the area behind the turbine in the averaged flow to compute the effect on power for turbines across a wide range of locations. This method allows for inspection of change in power for a wide variety of turbine array configurations. For comparison, this process can be repeated directly in FLORIS to compute the power of a downstream turbine for a range of locations.

   One advantage of using this method to compare wake predictions between models is that it focuses on the comparison on
the quantity of interest, which is the power production of turbines at a given location. Rather than trying to identify a wake center, focus is shifted from how far the centroid of deficit is shifted, to how much expected power production is possible at a given location. This will be important, if for example, focus shifts from wake deflection to energy entrainment.

   All simulations in this paper are of a neutral atmospheric boundary layer, with a mean-wind speed at hub height of 8 m/s, similar to what has been used in past studies (Fleming et al. (2015)). This simulation had at hub height 6% turbulence intensity
with a shear exponent of 0.085. The domain size is 5 km x 1.8 km x 1 km. The simulations include National Renewable Energy Laboratory's (NREL's) 5-MW reference turbines from Jonkman et al. (2009), modeled as an actuator disk with rotation for computational efficiency. In previous work, the actuator disk model with rotation has shown to be comparable to an actuator line model in predicting power (c.f. Porté-Agel et al. (2011); Martínez-Tossas et al. (2015))

## 3 One-turbine case

The one turbine case shown in Fig. 1 is the first case to be analyzed where a single NREL 5-MW turbine is placed in the flow. In one case, it is has no yaw offset (denoted as "baseline"), and then +/- 25°of yaw. In the first case, the hypothetical turbine is swept downstream with no lateral offset. The results are shown in Fig. 2.

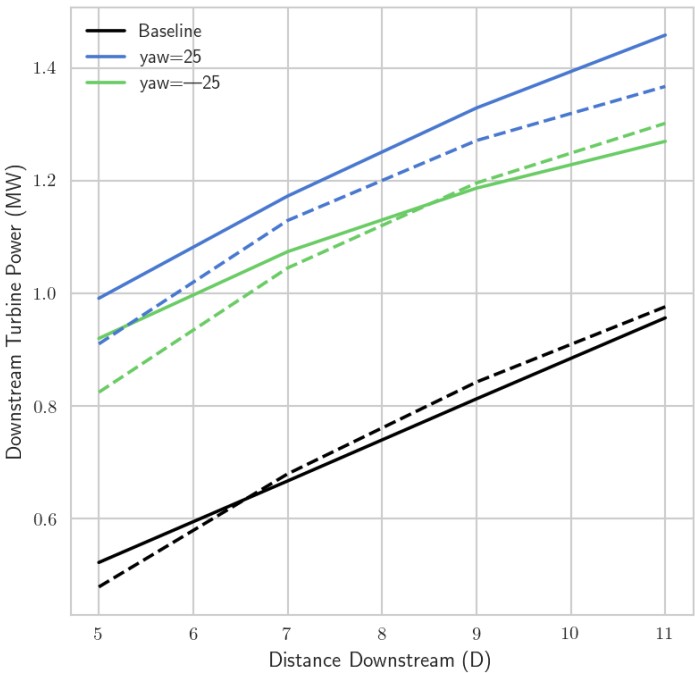

**Figure 2.** Power output in SOWFA (solid) and FLORIS (dashed) for a hypothetical turbine directly downstream of a baseline, or yawed turbine at different distances.

5   In both SOWFA and FLORIS, power is improved by misaligning the upstream turbine. In addition, more power is generated in the positive (CCW) yawed case than in the negative (CW) yawed case (see Fig. 2.) This asymmetry of wake steering, which has been documented previously, is an important result. For example, in the simulation study in Fleming et al. (2015), where two 5MW turbines are aligned directly in the flow, after deducting the power lost because of yaw on the first turbine, only positive yaw produces an overall gain in power for a case of two turbines separated by 7 diameters (note this wouldn't be true
10   for example in certain partial overlap cases, but it demonstrates the important of this asymmetry in wake steering).

In the original FLORIS model (Gebraad et al. (2016),) this asymmetry is captured by assuming that a wake has a certain natural deflection, even when operating in non-yawed conditions. This offset was previously used to describe the asymmetry

between positive and negative wake-steering angles. Conventionally, this natural offset angle was $0.13°$. This paper further reconsiders the source of this asymmetric behavior of the wake and its impact on the wind farm control strategy.

One way we can investigate the impact of the generated vortices on the wake is to look at cut-through slices of the flow at a distance downstream of the turbine. This is shown in Fig. 3. These cut-through slices are cut cross-wise through the flow direction, and include the average value of all velocity components. In these plots, the time-averaged flow with no turbines is first subtracted and so only the change from the background is plotted. These cut-throughs are plotted for both FLORIS and SOWFA.

Comparing the plots in the figure, we can note that FLORIS shows the impact of positive yaw as primarily impacting the location of the wake and the depth of the deficit. SOWFA, however, includes a large impact on wake shape, which is caused by the two counter-rotating vortices seen in Fig. 3. This is in line with the analysis of Vollmer et al. (2016); Howland et al. (2016).

As a first qualitative observation of the baseline cases, there does not appear to be any natural deflection (this will be further examined later). The natural deflection angle appears to be absent in this case and similar cases where the amount of veer in the simulation is kept to a minimum. Second, observing the in-plane flow, shown by arrows in the SOWFA plots in Fig. 3, the asymmetry in wake behavior likely comes from the interaction of the counter-rotating vortices with the wake (as seen in the SOWFA baseline case of Fig. 3) and the wind shear. More specifically, in the positive yaw case, the top vortex rotates constructively with the deflection of the wake and rotation of the wake (generated by the rotation of the turbine). In the negative yaw case, the lower vortex interacts more substantially with the rotation of the wake. The top vortex is also stronger because the wind speed is stronger at the top of the rotor as compared to the bottom of the rotor due to the shear layer and the interaction with the ground. This difference in vortex interaction may explain the asymmetry in the wake based on yaw misalignment. A third observation is that, in the positive yaw case (Fig. 3), the left side of the wake has been moved further to the right than FLORIS estimates, which computes deflection based on the center alone.

Fig. 4 shows the power of the hypothetical turbine that is swept laterally across the wake at different distances downstream. The left column of figures show the wake "profile" (measured in MW), and we see very good agreement between SOWFA and FLORIS. Note that the largest impact in gain in power (right column) is offset from turbine directly downstream. This percent gain includes the loss of the upstream turbine.

Under positively offset yawed conditions, the power improvements for turbines negatively offset from the wake center line are greater than what FLORIS estimates for distances greater of 7 D and more.

We note here that this effect cannot be modeled through a re-tuning of FLORIS. The main parameter available for implementing deflection asymmetry is a natural deflection angle. However, observing the "profile" of the baseline SOWFA case, it is not deflected, it is centered about zero. Increasing the natural deflection angle in FLORIS would raise the error observed in the baseline (as well as the negatively yawed case.)

Further, we cannot simply assume the wake recovers more quickly in positively yawed cases, because we note that in all cases, the depth of the trough of the deficit is predicted accurately by FLORIS. Therefore, the shape of the wake is a factor in being able to more accurately predict the power gain due to yaw misalignment, especially in cases of turbines offset from exactly downstream.

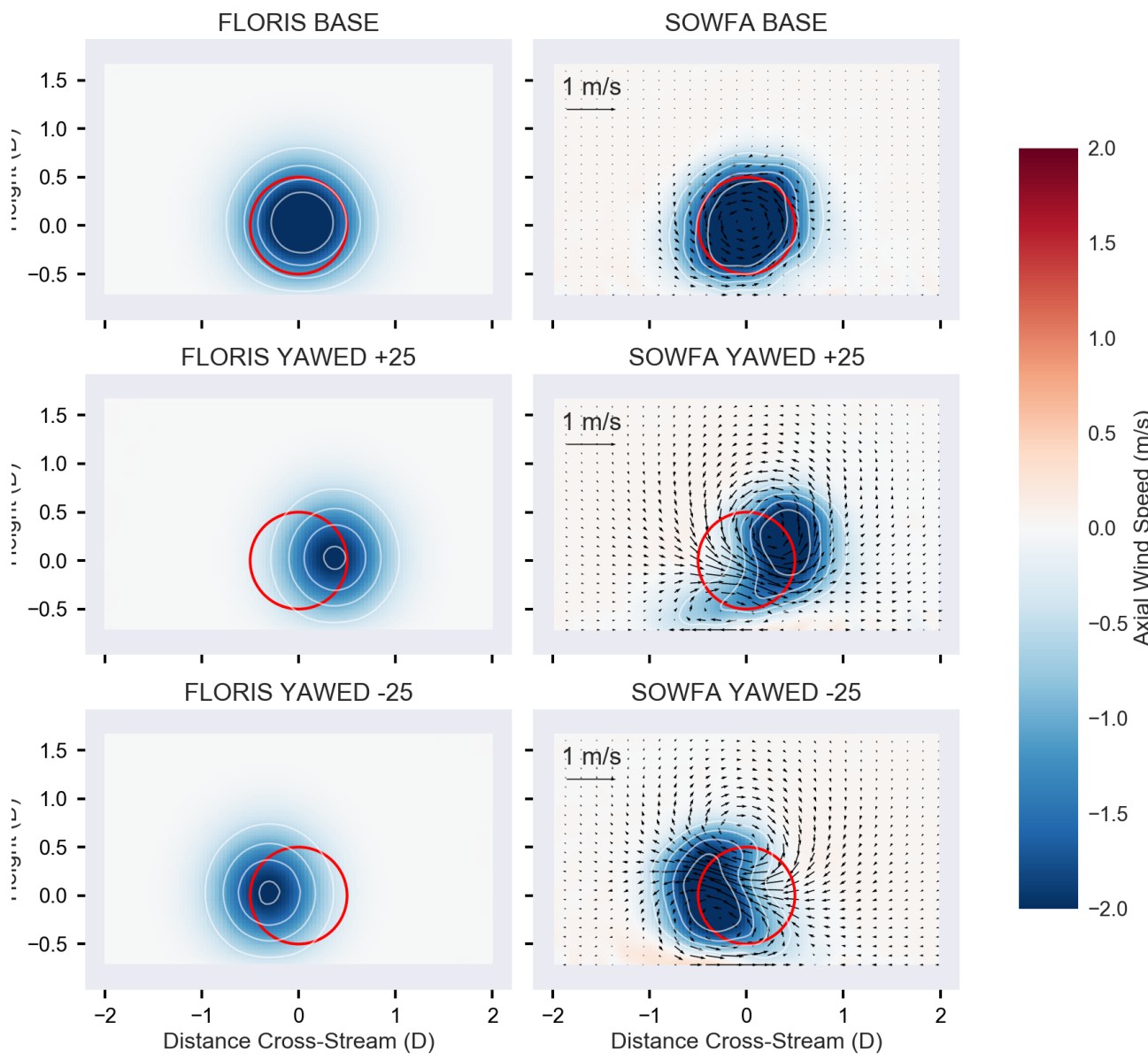

**Figure 3.** Cut-through of SOWFA flow (viewed from upstream), minus the background flow, at a distance of 7 diameters behind the turbine for baseline and yawed 25° and -25°. Deficit in axial flow is shown in blue, saturating at a deficit of 2 m/s to broadly show the wake area. For the SOWFA cases, the in-plane flow is visualized by arrows whose relative size indicates the strength of the flow. Red circles indicate a rotor located directly downstream of the turbine.

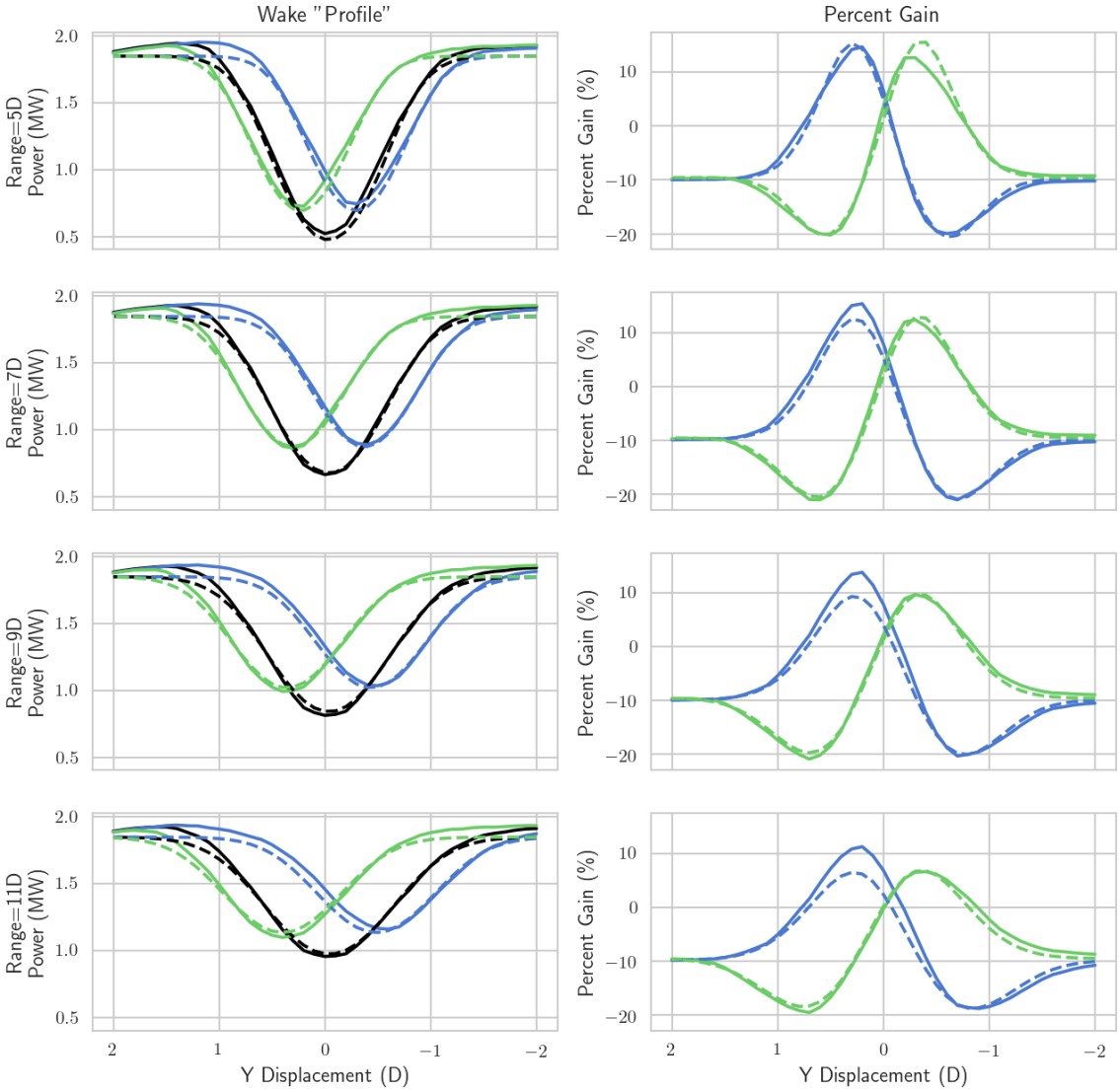

**Figure 4.** Power output of hypothetical downstream turbine for various downstream distances and lateral offsets (left). Solid lines are SOWFA results, and dashed are from FLORIS. On the right, the percent gain (of the total power of the actual and hypothetical turbine) is shown relative to the baseline of the two yaw angles.

A first result of the paper is the suggestion that the asymmetry in wake steering, where positive yaw is superior to negative yaw, is better explained by vortex-wake interactions than a natural deflection angle. This is more important for a two-turbine interaction when the downstream turbine is not directly downstream.

## 4 Two-turbine case

The impact of the counter-rotating vortices becomes even more clear when the simulation includes multiple turbines. In a second simulation, a case is considered where two turbines are in the flow 7 diameters (D) apart. The first turbine is either operating with no yaw misalignment or operating with 25° yaw misalignment while the second turbine is always aligned. The layout is shown in Fig. 5.

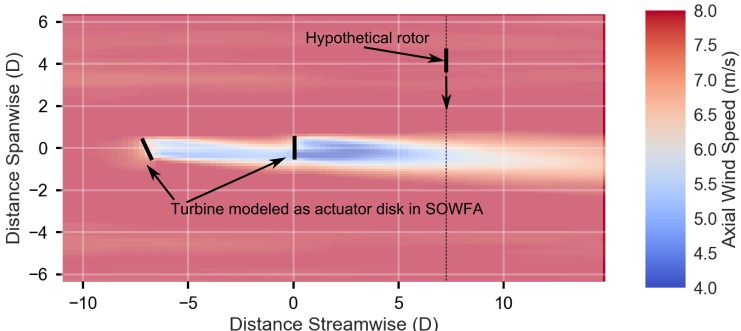

**Figure 5.** Layout of SOWFA simulation as in Fig. 1, now with two turbines. Note that only the first turbine is yawed in controlled cases.

Horizontal cut-throughs of the flow at hub height in SOWFA and FLORIS are compared in Fig. 6. The most striking obser-
10 vation is that in the SOWFA case, the second turbine's wake appears to be deflected, even though that turbine is aligned to the flow. In other words, the second wake appears to also be steered. We will refer to this phenomena as "secondary steering". This is not predicted by FLORIS. If we observe a cut-through at 7 diameters downstream of the second turbine, we see that there is a deflection of the wake of the second non-yawed turbine, see Fig. 7.

Combining observations from Fig. 6 and Fig. 7, we observe that FLORIS expects the wake of the second turbine to be
impacted by the first wake primarily through reduced inflow velocity and increased turbulence. Both of these effects do not lead to the deflection of the second wake. In this case, the second wake is significantly deflected in SOWFA. This deflection appears to be explained as a combination of the rotating wake of the second turbine, with the top vortex generated by the first yawed turbine. Considering the impact on power of a hypothetical third turbine defined 7 diameters downstream of the second, the difference between FLORIS and SOWFA is substantial. As seen in Fig. 8, FLORIS expects a maximum gain for
the three-turbine total of approximately 7%, where SOWFA predicts a total power gain of 17%. Also, SOWFA finds some locations where the 3-turbine array loses power, while FLORIS does not.

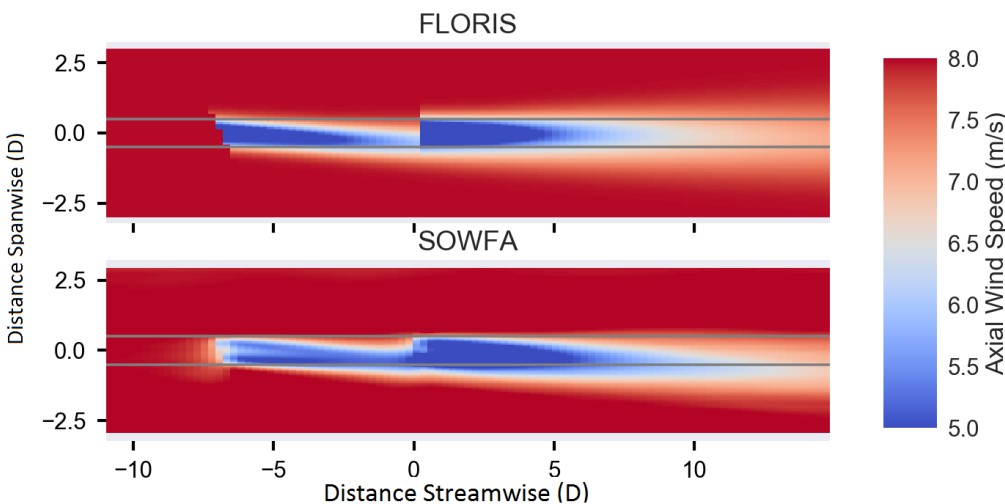

**Figure 6.** Horizontal cut-through of two-turbine flow in FLORIS and SOWFA. Gray lines pass through edges of rotor to help distinguish flow location relative to rotor.

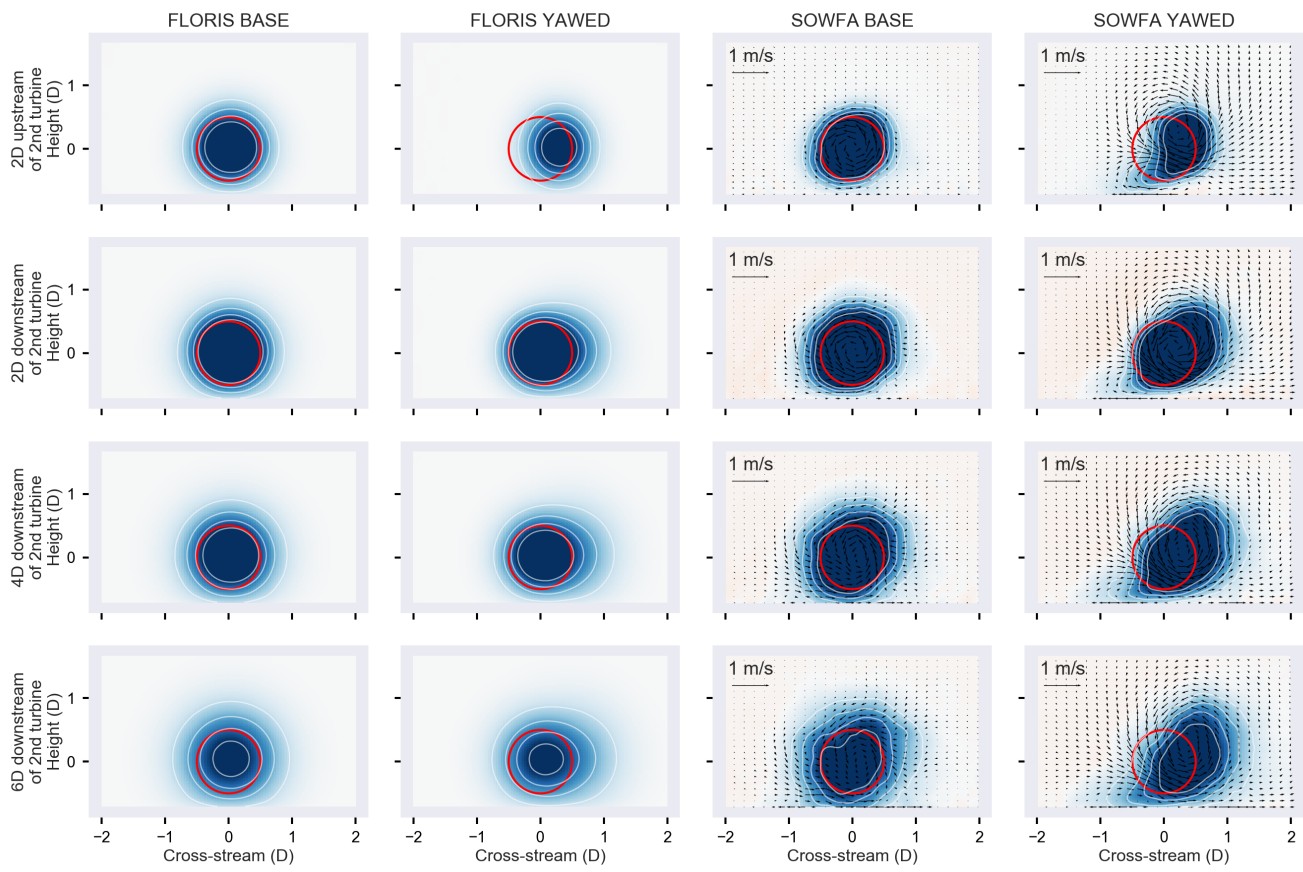

**Figure 7.** Cut-through visualizations of FLORIS and SOWFA 2D in front of second turbine (top row) and behind the second turbine (remaining rows).

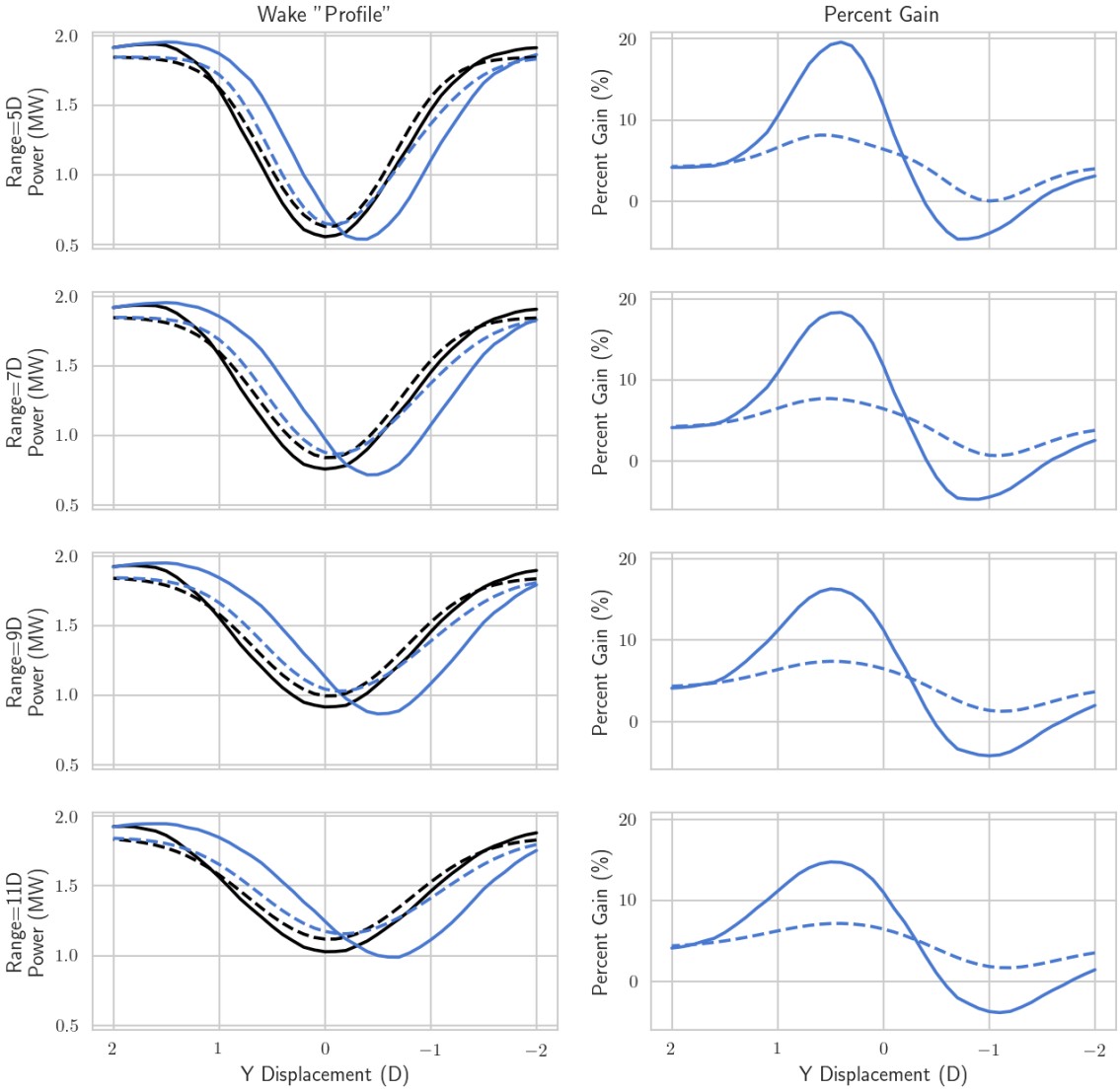

**Figure 8.** Power of a hypothetical turbine behind the 2nd turbine in SOWFA (solid) and FLORIS (dashed) (left) and percent increase relative to baseline (right). FLORIS has good agreement in the baseline case but misses the change in wake behavior obtained in the yawed case in SOWFA.

## 4.1 Discussion of secondary steering

In the previous section, the concept of "secondary steering" is introduced, wherein, a steered wake, causes a deflection of the wake of a downstream turbine, that is not itself yawed. This has important implications for wind farm controller design, and it is critical to understand the driving mechanisms.

In this section, we consider an alternative explanation for the observed phenomena and discrepancy with the FLORIS engineering model. In the previous section, the vortices generated by wake steering are suggested as the cause of the "secondary steering" phenomena, because they can be seen in flow analysis to generate persistent cross-flows that continue after the combination of wake with the second turbine.

However, a possible alternative explanation is that the first deflected wake, is producing a partial wake scenario. FLORIS' wake combinations model, as described earlier, is based on sum of squares. Given that Trabucchi et al. (2017) shows that sum-of-squares superposition, as is used in FLORIS, has an error in prediction relative to newer methods, this could be the explanation for the discrepancy, i.e., partial wake overlap leads to an apparent secondary steering and improved wake combination models in FLORIS would resolve the discrepancy.

To test this, a new SOWFA simulation is run, similar to the two-turbine cases described in the previous section and shown in Fig. 6. However, in this case, the front turbine is moved down $1/4D$. In this configuration the wake of the upstream turbine overlaps the downstream turbine in a similar location as the steered wake of the originally located turbine. However in this case the generated vortices are not produced.

Fig. 9 shows the results of this new comparison. On the left column, the wake "profile" $2\,D$ upstream of the downstream turbine is shown in both FLORIS and SOWFA. The "partial" wake case shows a deficit in a similar location to the yawed case. In the right column are the "profiles" from FLORIS and SOWFA $14\,D$ after the downstream turbine. In the upper FLORIS figure, you can see that FLORIS assumes that both yawing and translating the front turbines will cause the power deficit far downstream to be both shifted a little bit, and in a similar way. However, SOWFA shows that while the FLORIS prediction of the combined partial wake is not as accurate as it's prediction of the incoming single wake (as could be expected from Trabucchi et al. (2017)), the substantial change in the yawed case is much more different than FLORIS's prediction. Note for example the minimum of the "profile" of the yawed wake far downstream is further deflected than the incoming steered wake. This is the "secondary steering" and probably can't be modeled as combination of partial overlap only, as in that case the minimum should appear in between the minimum of the two wakes to be combined.

## 5 Multiple-turbine case

Building on the results from the three-turbine case, a final case simulates a tightly spaced wind farm and is used to further explore the importance of these counter-rotating vortices generated under yawed conditions. A 12-turbine wind farm, shown in Fig. 10, is modeled in both SOWFA and FLORIS. The case uses the same inflow, as well as runtime (2000 s) and averaging time (1600 s) of all previous cases.

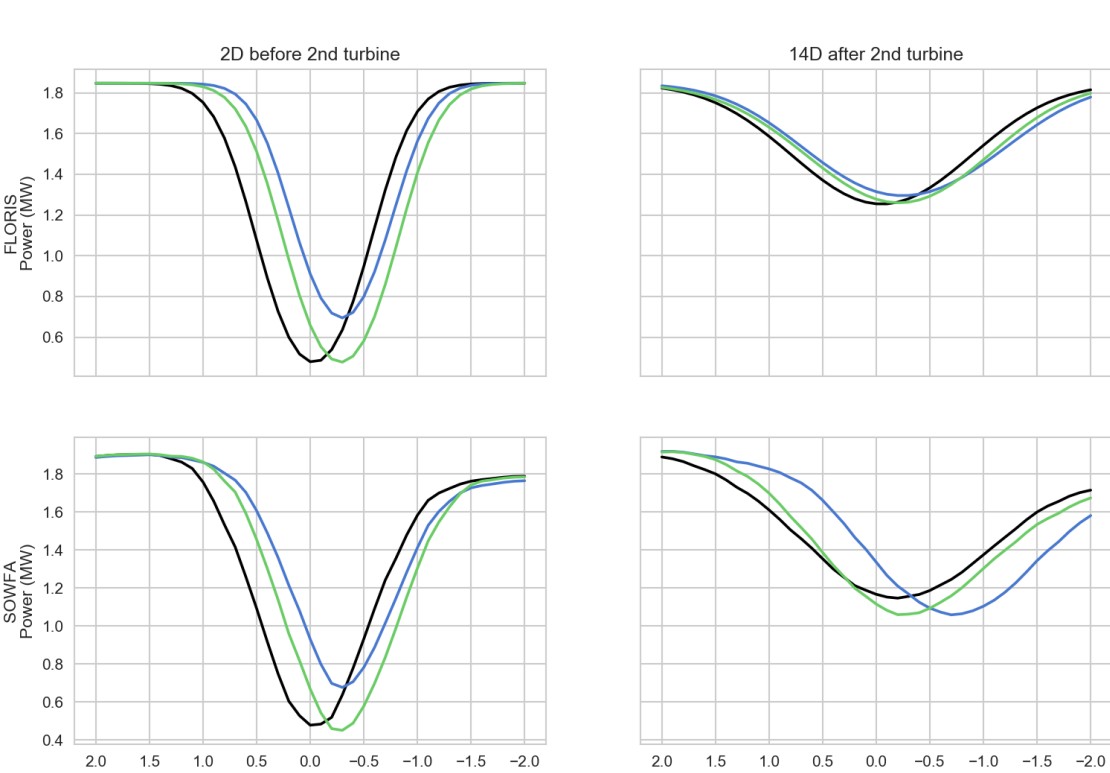

**Figure 9.** Power of a hypothetical turbine in Baseline, Yawed (+25°), and partial wake. The left column shows the power profile 2D before the 2nd turbine, while the second column is 14D behind.

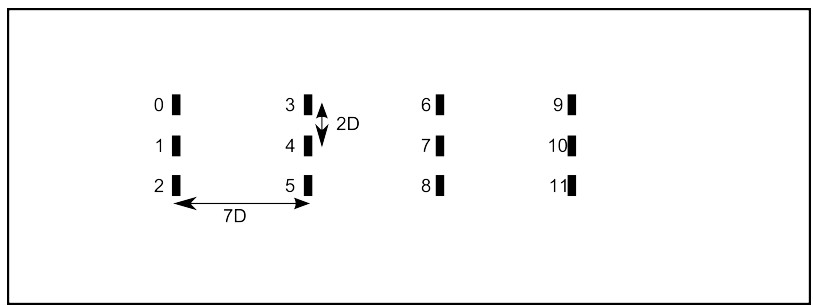

**Figure 10.** Layout and numbering of turbines in a turbine array case for both SOWFA and FLORIS.

This layout affords the opportunity to assess how the consideration of vortices impacts previous assumptions of wake steering. If wake steering only deflects wakes to the right or the left of a downstream turbine, then it is possible to incidentally steer a wake into a different turbine operating nearby. This description suggests that wake steering will be difficult in situations where a turbine's wake cannot be directed easily into empty space.

5    However, noting that vortices and wakes combine in a profitable manner when wakes interacted going downstream, we can speculate that vortices can provide a constructive mechanism for wakes located near to each other laterally as well.

Additionally, in FLORIS as currently modeled, it is expected that although the three horizontal rows of turbines are tightly spaced, the effect of simultaneously applying wake steering to turbines 0, 1, and 2, would be equivalent to the super-position of applying wake steering to each row individually. However, in SOWFA, the generation of these large-scale structures, i.e.

10    counter-rotating vortices, influence each other and can combine in ways that are not captured in the current engineering model.

We ran five simulations of the scenario in Fig. 10, one in which all turbines operate aligned ("baseline"), three where only one of the upstream turbines is yawed ("yaw0," "yaw1," and "yaw2"), and a final where all three are yawed ("yawAll"). Then in the analysis, we can compare the results of summing the effects of individually yawed turbines from separate simulations ("sumInd") in post-processing to the case of simultaneously yawing all turbines within the simulation ("yawAll").

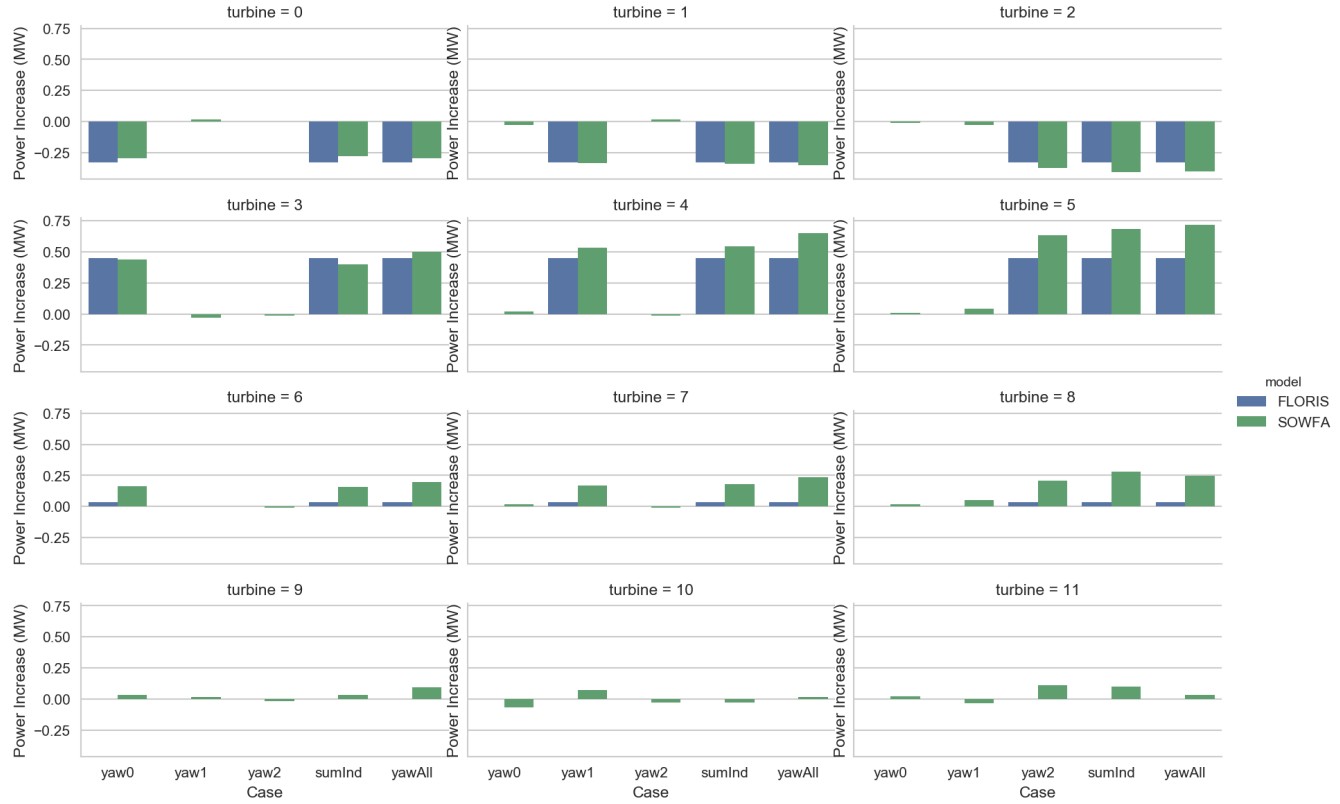

**Figure 11.** Change in turbine power relative to baseline for each turbine across the various scenarios in both SOWFA and FLORIS.

Fig. 11 shows the change in power from baseline, for each turbine, for SOWFA and FLORIS. We note that the power predicted for the baseline case is 13.05 MW for FLORIS, and 13.107 MW for SOWFA, which indicates good agreement.

For the first row, which is the upstream turbines implementing the yaw misalignment, the losses between SOWFA and FLORIS are similar, and small differences between individual and simultaneous yawing is observed. However, in the second row, while FLORIS continues to show no difference between individually and simultaneously yawed conditions, SOWFA shows a consistent gain when yawing is simultaneously applied versus summing of individual yaw misalignment effect ("yawInd".)

In the third row, as indicated in the earlier three-turbine analysis, FLORIS sees no impact on turbine power, whereas SOWFA observes an increase, and a difference between individual and simultaneous yawing. This continues to a lesser extent in the fourth row, now a full 21 diameters behind the yawed turbines.

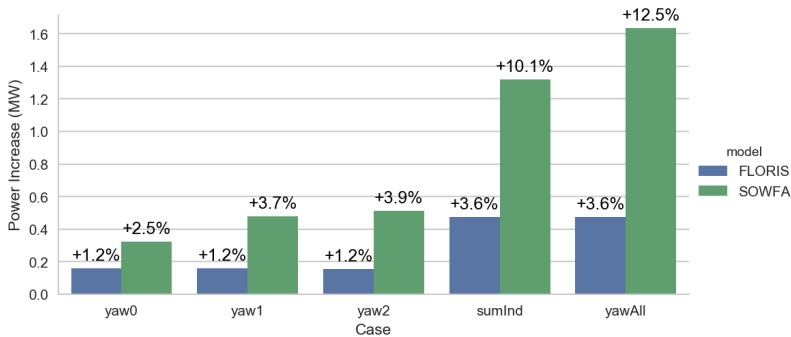

**Figure 12.** Change in total power for SOWFA and FLORIS across the cases.

The summed turbine power is shown in Fig. 12. Again, FLORIS underestimates the power increase from the later rows, and assumes that individually applied yaw offsets can be summed to the whole, while SOWFA shows an increase from simultaneous application. Fig. 13 shows cut-throughs following each of the rows in the "yawAll" simulation. Most notably, it appears that the counter-rotating vortices that are generated from the three upstream turbines are combining to have a larger impact on the downstream turbines. Just by operating the upstream turbines under misaligned conditions, these large-scale structures are generated and propagate throughout the farm.

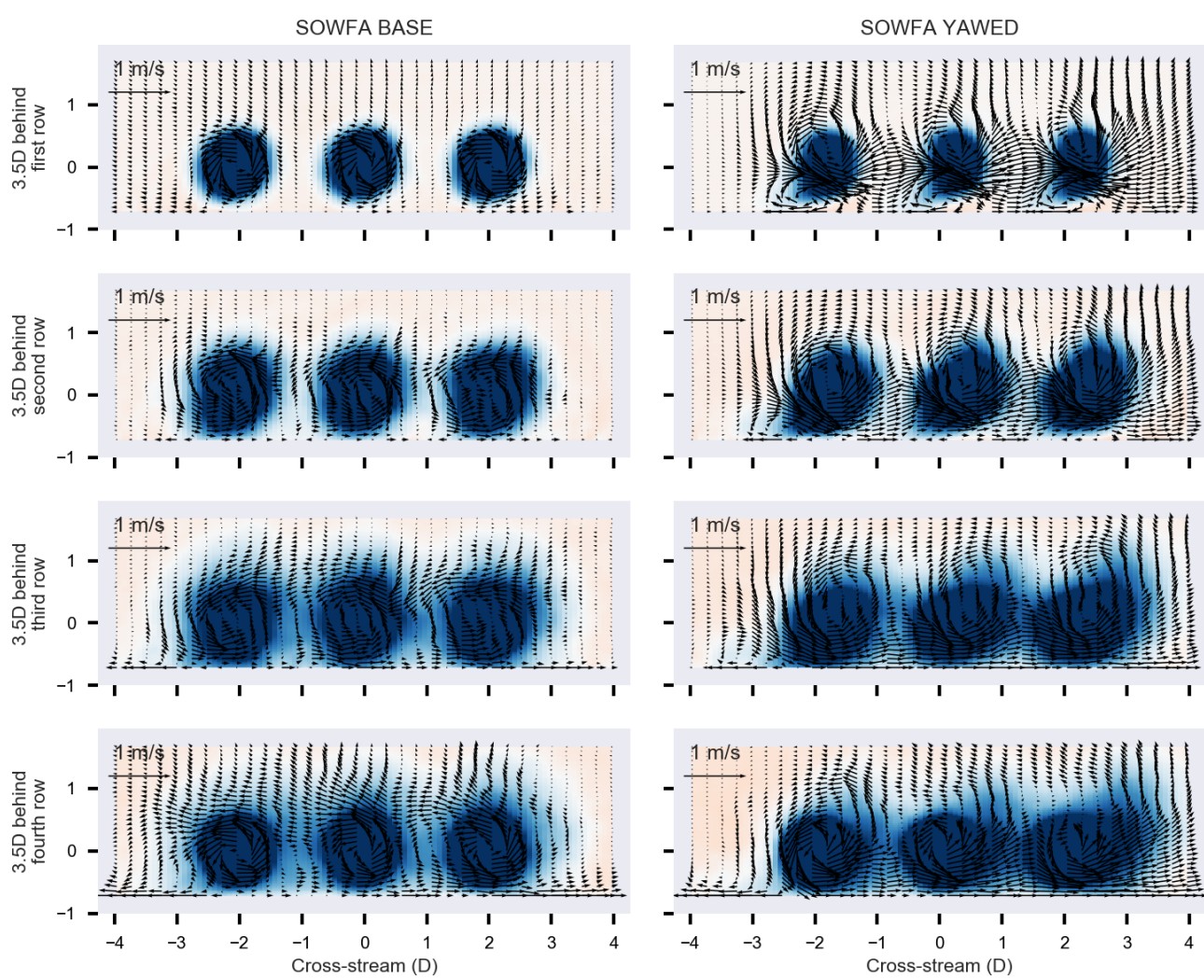

**Figure 13.** Cut-through visualizations of SOWFA following each row in the turbine array scenario "yawAll".

# 6 Conclusions

In this paper, the role of flow structures, particularly a pair of counter-rotating vortices, generated in wind farm control through yaw misalignment was investigated. These vortices were shown to be important in (1) deforming the shape of the wake, (2) in explaining the asymmetry of efficacy in wake steering of oppositely signed yaw angles (rather than the currently used natural deflection angle), and (3) in understanding how a "steered" wake interacts with a downstream wake and with laterally adjacent wakes. These findings are critical for developing an effective, robust strategy for wind farm control.

One important finding of the paper was that, because of the presence of these vortices, a steered wake can deflect the wake of a downstream turbine, even if that turbine is not yawed, a process called "secondary steering". A second finding was that combinations of wakes from yawed turbines are shown to involve merging of generated cross flows. These combinations can lead to changes in power generation that are different then adding the changes of yawing individual turbines.

In ongoing research at NREL, new engineering wake models which contain these behaviors due to the generated voritices are under development to be incorporated into FLORIS. Once completed these new models will be compared to similar SOWFA data sets to determine if they can resolve all discrepancies. Such new models, can then be used to develop new wind farm controllers. These new wind farm controllers can be optimized to take advantage of the controllability afforded by the vortices ability to manipulate the flow and will yield a more powerful form of wake steering than currently exists.

*Acknowledgements.* The U.S. Government retains and the publisher, by accepting the article for publication, acknowledges that the U.S. Government retains a nonexclusive, paid-up, irrevocable, worldwide license to publish or reproduce the published form of this work, or allow others to do so, for U.S. Government purposes.

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
