# Peer review of "A simulation study demonstrating the importance of large-scale trailing vortices in wake steering"

_Wind Energy Science, 2017_

## Referee Comment (RC1) · Anonymous Referee #1 · 11 Jan 2018

The paper "From wake steering to flow control" by Fleming et al. is generally well-structured and yaw-based wake deflection is a topic of high interest. Three different wind farms are simulated in both the LES model SOWFA and the engineering wake model FLORIS. The results show that the baseline cases agree well, while the application of yaw-based wake deflection at the front-most turbine introduces a deviation between the models. This deviation increases after the second turbine in a row. The authors relate the difference between the models to the cross-stream component of the flow, that is not included in the FLORIS model. Specifically, they observe two counter-rotating vortices in the wake flow of the yawed turbine, that are regarded to influence the wakes of downwind turbines. In an LES of a larger wind farm they observe that the power output of a simultaneous control of all turbine rows differs from the power addition of individually controlled turbine rows. They conclude in claiming that a new concept of flow control could make use of the observed effects.

[Figure]

Technically the analysis of the simulation studies is good, the chosen visualizations mostly allow to detect the differences between the two models.

Unfortunately, the very short description of the models makes it difficult to fully believe the argumentation of the authors.

The authors might overstate the influence of the observed vortices on the difference between LES-model and engineering model. An analysis or discussion of other potential sources, e.g. wake superposition, turbine model, flow entrainment, is missing.

Title and abstract of the paper are promising more than the paper actually delivers, as no new concept of flow control is developed or presented, but is only suggested.

These major aspects need to be addressed before we can recommend it for further publication in the Wind Energy Science journal.

Specific Comments:

1. Title and abstract: Please relate the title and abstract more to the content of the paper. Also, the term "new physics" should be replaced by a more specific term.

2. P01. L26: Since the focus of study is on the interaction of multiple wakes, it is necessary to briefly introduce how multiple wakes are modeled in FLORIS and what is missing. A more detailed description of FLORIS would fit better into section 2.

3. P04. L09: Besides a mean wind speed of 8 m/s (hub height?) no further information on the inflow condition are given. Similar to past studies? To what extent? This part should be improved by adding further inflow characteristics (e.g. inflow profiles, TI, ...)

4. P05. L08: In the paper of Fleming 2015 this has only been investigated for the specific case that the upstream turbine is yawed and the downstream turbine has no lateral offset. Can you conclude that there are no cases where a negative yaw offset would be beneficial?

5. P07. L03: "there does not appear to be any deflection" -> Due the colorbar it is

impossible to see any details of the flow structure. It indeed looks like no deflection of the wake took place. But what is YOUR definition of wake? Could some wake tracking be applied in order to make a reliable and clearer statements?

6. P10. Figure 7: It seems that the employed FLORIS model only considers the wake deficit for modeling multiple wakes and does not consider the superpositions of deflected wakes by upwind turbines. Since FLORIS benefits from the generic AD model, it might be good to be evaluated with a LES model with the non-rotating AD model as well. Moreover, conducting such observations provides a clear view on the physics of the counter-rotating vortices.

7. P10. L10: Please also discuss the other potential sources for the deviations between FLORIS and SOWFA. While the baseline cases indeed show a good agreement, wake superposition is a potential source of error, e.g. wake superposition, turbine model, flow entrainment, especially for half-wake conditions (see e.g. Machefaux, "Multiple wakes", 2015 or Trabucchi et al., "3-D shear-layer model for the simulation of multiple wind turbine wakes: description and first assessment", 2017), which is essentially what is happening when the wake of the upwind turbine is deflected.

8. How could the FLORIS model be potentially improved? Or which further analysis are required before such recommendations could be substantiated?

Technical Corrections:

9. P01. L10: The first part should include a few more literature references. The literature review on engineering models is limited just like the one about the interaction of multiple wake. Furthermore, there have been early wake deflection studies in the wind tunnel which might be worth mentioning.

10. P01. L13: "Early work in wake steering …" This sentence somehow suggests that first investigations of wake steering were performed using computational fluid dynamics. In the years before, however, multiple investigations were carried out in the wind

tunnel. Please consider rephrasing this part.

11. P01. L22: "The model is shown to predict the behavior of wake steering and …". If the model does predict the behavior of wake steering why do we need other models? What are the weaknesses of the model? Consider adding further details or rephrase the statement.

12. P02. L16: Typo: "to changing a the deficit"

13. P03. L01: The 1st and 2nd objectives of the paper seem similar. They might be combined or rephrased.

14. P03. L05: Some brief description of the FLORIS model, especially the handling of multiple wakes and the assumed natural wake deflection, may help a reader for following the comparative studies.

15. P03. L03 and L19: The 3rd objective is not addressed enough through the paper. The described approach of a hypothetical wind turbine downstream (in section 3) is introduced for reducing computational efforts of the simulation studies. It might be also good to discuss the benefits of this method for wind farm control research.

16. P03. L06: "models of wind farms" -> wind farm models?

17. P03. L06: Who developed the two models? References

18. P03. L16: It seems that the actuator disc with rotation is used in the SOWFA simulations. It should be clarified.

19. P03. L18: ".Churchfield et al." Reference Typo

20. P03. L20: " the two models have been used" -> FLORIS and SOWFA? There is no real separation between Chapter 2 "Models" and Chapter 3 "Method". This seems rather unusual.

21. P03. L20: In the 1st paragraph, some references for past research would be

beneficial.

22. P03. L27: "( see Fig 1.)" Typo, unnecessary space before "see".

23. P03. L28: Why averaging the flow for 1600s? What is the reason for this specific amount of time?

24. P03. L29: "by an additional hypothetical at some point" Typo, "turbine" missing?

25. Figure 1: Colorbar?

26. Figure 1: Without a grid or ticks the use of the tick labels is limited

27. Figure 1: View from above down at the plants?

28. Figure 1: Consider including a sketch to define positive and negative yaw offsets (CWW, CW)

29. P04. L01: How do you determine the averaged flow?

30. P04. L04: Reference for FAST

31. P04. L12: "this method has shown to be comparable to an actuator line model in predicting power" -> To what extent?

32. P05. L03: "+/- 25âŮę" The degree sign does not show up right in the reviewed draft. Please verify if this is a typo.

33. P04. L04: Please consider adding the coordinate system to Figure 1

34. P05. L07: "For example, in Fleming ..." Please consider rephrasing this sentence due to its complex structure.

35. P04. L27: Please distinguish between the approach of the current paper and the approach of the prior studies in the previous paragraph.

36. P05. L12: "However, this natural deflection in non-yawed conditions is not observed in simulations without the Coriolis effect ..." -> add reference

37. Figure 3: It should be clearly stated whether the wake flow is looked at from upstream or downstream

38. Figure 3: The color scheme should be avoided since information is lost when printed in greyscale.

39. Figure 3: Why are the limits of the colorbar -2 m/s to 2 m/s. Are there any positive values at all?

40. Figure 3: Grid lines or ticks should be added

41. Figure 3 and Figure 4: Merging both figures would certainly make it easier for the reader to compare the impact of a positive and negative yaw misalignment on the shape of the wake.

42. Figures 3,4,8,13: Please add a reference scale for the arrows to be able to see the magnitude of the vortices.

43. P06. 03: "cut-through slices of the flow at a distance downstream of the turbine." -> flow = longitudinal velocity component?, vertical slice?

44. P06. 08: The analysis was performed with a AD model of the turbine. The results look like the rotation of the rotor is included. Often this is not the case if an AD implementation is used. Therefore, this should be clearly stated when the model is introduced whether the rotation of the rotor is included or not. In case it is not included it should be commented on: To what extent could the results differ in a model that takes rotation into account? Is the use of "only" an AD model justified for this type of investigation.

45. Figure 4: Same comments as for Figure 3

46. P07. L02: "Visual inspection indicates some discrepancies . . ." -> In the following sentence an example about a similarity is given. Consider rephrasing.

47. Figure 5: Caption: "showing good agreement . . ." -> Consider moving the analysis

to the main text.

48. Figure 5: yaxis labeling for right column missing

49. P09. L02: Mention the method used for obtaining the wake center

50. P09. L07: "Under positively ..." Should it be positively offset?

51. P09. L07: "Under positively ..." Should it be "7D and more" instead of 5D? No results are shown for distances greater than 5D and smaller than 7D.

52. P09. L07: What about negatively yawed conditions?

53. Figure 6: Same comments as for Figure 1 (greyscale of colorbar, ticks, grid, ...)

54. Figure 7: Same comments as for Figure 1 (greyscale of colorbar, ticks, grid, ...), add "at hub height" in the caption

55. Figure 8: Same comments as for Figure 3

56. Figure 8: Caption "visualization of FLORIS 2D", add SOWFA

57. Figure 8: YLabel: Everything is stated in "D". Here no explanation is given, the D is missing and X is introduced. Compare labeling with Figure 13

58. Figure 9: Same comments as for Figure 5

59. Figure 9: Caption: "Power of a hypothetical behind" -> Word missing "turbine"

60. P13. L02: "... three-turbine cases, A ..." Typo

61. P13. L02: "A final case simulates a tightly spaced wind farm is used to further explore ..." -> Grammar?

62. P13. L02: Please provide the averaging time of the SOWFA simulations

63. P13 L17: The statement "we can compare the result of summing the effects of ..." is not clear enough. It might be reformulated better.

64. P14 L3: Please specify that Fig.13 corresponds to "YawAll" scenario. Moreover, a bar graph helps to evaluate the results.

65. P15. L03: Is there a specific reason for choosing 3.5D?

66. P16. L14: Are you not influencing the wake generation and due to this the characteristics of the downstream flow. Therefore, the focus is still on controlling the wakes, because they are your tool to influence the entire flow.

---

## Referee Comment (RC2) · Anonymous Referee #2 · 23 Jan 2018

[a4paper]article  The paper compares a control-oriented model and detailed CFD simulations of wind turbine wakes.  While in the baseline case the models fit well, the models disagree, if one or more turbines are yawed.  The authors state at the end of the introduction that the paper's contributions are:

1. A demonstration that the control-oriented model cannot describe all important effect of multiple wakes.

2. A demonstration that the effect is especially critical for arrays of multiple turbines.

3. A proposed new approach of wind farm control based on the analysis.

4. Recommendations of improvements to control-oriented modeling.

In the conclusion, the results are summarized in:

1. Motivation to develop new physics for a control-oriented model for wind farm control which included the effect of counter-rotating vortices.

2. Suggestion that wind farm control should be more about generation of large-scale flow structures and not about redirection only.

**Overview**

Overall, the paper is well-written and includes very important findings for the wind turbine control community. Although the paper is already in a very good state, the descriptions of contributions and results could be improved. Further, some more details could help to understand the analysis.

**Descriptions of contributions and results**

1. The title is somehow misleading: It is clear after the reading why you chose the title (your result #2, the suggestion), but actually don't really "control the flow" but you analyze the effect of yawing wind turbines.

2. This brings me to another important point: you nicely describe new effects which are very important for wind farm control. This deserves from my opinion more credits. It is somehow included in your contribution #1 (demonstration that FLORIS cannot describe it), but might be good to highlight it more in a positive way.

3. In this way, your contributions #1 and #2 might be also be more specific to the new effects: i) A upwind wake steers the wake of a un-yawed turbine (see Figure 6 and 2nd and 3rd row in Figure 11) and ii) and the wakes of wind turbines in
the same row have interactions which are unneglectable (sumInd different from yawAll in Figure 11). The original contributions #1 and #2 are also overlapping.

4. The contributions #3 and #4 are very similar to your results #2 and #1. However, the finding and description of the new effects are missing in the listing of the results. Further, the points i) Suggestion: "we need to control the flow and not only steer the wakes" and ii) Motivation: "this requires new models" are important but still quite vague to be listed as the main results of a scientific paper.

5. Addressing a similar issue: Your structure is: 1. Introduction, 2. Models, 3. Methods, 4-6 results, 7 conclusions. Sections 4-6 could be renamed to be more consistent, e.g. "one turbine case", "two turbine case", and "Multiple turbine case". Or: "Results for two turbines", "results for three turbines", "results for a wind farm" or similar. The current section titles are a little bit confusing.

**Missing details**

Since a large portion of the paper is to compare FLORIS to SOWFA and to show the limits of it, it would be helpful to describe it in more details.

**Other minor issues**

1. The ° sometimes is not a superscript, e.g. 2nd line of Section 4 and caption Figure 3.

2. Figure 3: Linking the arrow size to a wind speed might be helpful to get a better feeling for the effects.

3. Figure 4: Period is missing at end of caption.

4. Figure 5 and 9: ylabel of 2nd column is missing. And maybe the plots look smoother when you use a higher discretization (now it seems to be 0.2 D).

5. Figure 8: ylabel missing. And plots are too small. Maybe you could limit it to -2 D and +7 D.

6. Figure 9: "turbine" missing behind "hypothetical".

7. Figure 10: The zero-indexing might be confusing. Why not turbine 1 to 12 and "yaw 1" to "yaw 3"?

---

## Author Comment (AC1) · 25 Jan 2018

Thank you to the reviewer for a thorough and very helpful review! Please see the attached for our responses to all comments.

Please also note the supplement to this comment:
https://www.wind-energ-sci-discuss.net/wes-2017-52/wes-2017-52-AC1-supplement.pdf
* * *

---

## Author Response (AR1)

**Response To Review 1:**

Thank you for a very thorough review with many good suggestions! We have tried to incorporate all suggestions and document the changes. In some places we have added new analysis, and we have revised all sections to improve the focus of the paper in accord with your comments.

We have organized the changes made by topic in this document (but have kept your numbering for reference). All responses are in red, and following this, a version of the paper with changes highlighted is attached so all changes can be easily identified.

**Clarifying title/abstract/purpose/results**

*Title and abstract of the paper are promising more than the paper actually delivers, as*

*no new concept of flow control is developed or presented, but is only suggested.*

*1. Title and abstract: Please relate the title and abstract more to the content of the paper*

We agree with this critique. After re-thinking the major objective of this paper, the title has been changed to "A simulation study demonstrating the importance of large-scale trailing vortices in wake steering". The main goal of this paper is to demonstrate, through LES simulation, how vortices are important to understanding wake steering. This change has likewise caused changes in language to the abstract, introduction and conclusion.

*Also, the term "new physics" should be replaced by a more specific term.*

- This term is removed everywhere, and more specific language employed

**Additional model details**

**FLORIS**

*2. P01. L26: Since the focus of study is on the interaction of multiple wakes, it is*

*necessary to briefly introduce how multiple wakes are modeled in FLORIS and what is*

*missing. A more detailed description of FLORIS would fit better into section 2.*

- This is now included and additional references are provided. Here is the new text on FLORIS:

*This paper focuses on two specific wind farm models. First, FLORIS, is a low-fidelity, control-oriented tool for wind farm control, which includes several possible wake models developed by NREL and TU Delft*

*Gebraad et al. (2016). FLORIS is python-based, open source, and available for download on github (https://github.com/WISDEM/FLORIS). The overall approach of FLORIS is to provide an engineering model of wakes that predicts the important average behaviors of wakes in a computationally efficient way such that it can be used to derive control strategies through optimization or function as an internal controller model. A recent report compares predictions of the latest FLORIS model to lidar data from a utility-scale wind turbine operating in yaw and provides good agreement (Fleming et al. (2017a)).*

*In this work, the wake model used in this paper assumes a Gaussian wake that is derived from self-similar turbulence theory (Bastankhah and Porté-Agel (2014, 2016); Niayifar and Porté-Agel (2015)). The wake expands linearly and the parameters of this Gaussian wake are a function of ambient turbulence intensity and turbine operation. Overlapping wakes are combined using a sum-of-squares approach that has been used previously in literature (Kati´c et al.). It is important to note that there are alternative methods to combine wakes as wake superposition is an ongoing research topic (Machefaux and Mann (2015), Trabucchi et al. (2017)).*

*Wake deflection is also included in this model based on the yaw misalignment of a turbine. It is modeled using a budget analysis of the Reynolds Averaged Navier-Stokes equations. For further details, the reader is referred to (Bastankhah and Porté-Agel (2016)).*

*14. P03. L05: Some brief description of the FLORIS model, especially the handling*

*of multiple wakes and the assumed natural wake deflection, may help a reader for*

*following the comparative studies.*

- Description of FLORIS model is expanded as above, and the assumed natural deflection angle Is specified in the one turbine section.

*20. P03. L20: " the two models have been used" -> FLORIS and SOWFA? There is*

*no real separation between Chapter 2 "Models" and Chapter 3 "Method". This seems*

*rather unusual.*

- This has been clarified and the models and methods chapters are combined

**SOWFA**

**3. P04. L09: Besides a mean wind speed of 8 m/s (hub height?) no further information**

**on the inflow condition are given. Similar to past studies? To what extent? This part**

**should be improved by adding further inflow characteristics (e.g. inflow profiles, TI, . . .)**

- Paragraph has added details:
- *All simulations in this paper are of a neutral atmospheric boundary layer, with a mean-wind speed at hub height of 8 m/s, similar to what has been used in past studies (Fleming et al.*

*(2015)). This simulation had 6% turbulence intensity with a shear exponent of 0.085. The domain size is 5 km x 1.8 km x 1 km. The simulations include National Renewable Energy Laboratory's (NREL's) 5-MW reference turbines from Jonkman et al. (2009), modeled as an actuator disk for computational efficiency. In previous work, the actuator disk model with rotation has shown to be comparable to an actuator line model in predicting power*
*(c.f. Porté-Agel et al. (2011); Martínez-Tossas et al. (2015))*

*23. P03. L28: Why averaging the flow for 1600s? What is the reason for this specific*

*amount of time?*

- The goal was to average as much time as possible, given the computational time limits of our computing system (two days of running time allowed per job on most queues).  The first 400 seconds are removed by calculating the time a wake would take to propagate through the area of analysis.

*29. P04. L01: How do you determine the averaged flow?*

- The following sentence is added: The averaged flow is provided directly via OpenFOAM output functions.

*31. P04. L12: "this method has shown to be comparable to an actuator line model in*

*predicting power" -> To what extent?*

- We've added two references to this sentence:
- In previous work, the actuator disk model with rotation has shown to be comparable to an actuator line model in predicting power (c.f Porte-Agel et al (2011); MArtinez-Tossas et al (2015))

*36. P05. L12: "However, this natural deflection in non-yawed conditions is not observed*

*in simulations without the Coriolis effect . . ." -> add reference*

- This sentence was based on internal, but not yet published results, and we have decided to remove it so that we can further validate before stating in a publication

*62. P13. L02: Please provide the averaging time of the SOWFA simulations*

- This is now included

*18. P03. L16: It seems that the actuator disc with rotation is used in the SOWFA*

*simulations. It should be clarified.*

- This is now stated explicitly in this sentence: In this approach, the SOWFA simulation includes one turbine modeled as an actuator disk with rotation

**Wake combinations**

*Technically the analysis of the simulation studies is good, the chosen visualizations mostly allow to detect the differences between the two models.*

*Unfortunately, the very short description of the models makes it difficult to fully believe the argumentation of the authors.*

*The authors might overstate the influence of the observed vortices on the difference between LES-model and engineering model. An analysis or discussion of other potential sources, e.g. wake superposition, turbine model, flow entrainment, is missing.*

*7. P10. L10: Please also discuss the other potential sources for the deviations between FLORIS and SOWFA. While the baseline cases indeed show a good agreement, wake superposition is a potential source of error, e.g. wake superposition, turbine model, flow entrainment, especially for half-wake conditions (see e.g. Machefaux, "Multiple wakes", 2015 or Trabucchi et al., "3-D shear-layer model for the simulation of multiple wind turbine wakes: description and first assessment", 2017), which is essentially what is happening when the wake of the upwind turbine is deflected.*

- This is a really excellent point and we thank the reviewer for the suggestion.  We performed some additional simulations to test these suggestions, and based on those results added an entirely new section titled  "discussion of secondary steering".  The section includes a new figure which compares a yawed two-turbine scenario with a partial-wake (but not yawed) scenario.

**Actuator Disk Model**

*6. P10. Figure 7: It seems that the employed FLORIS model only considers the wake deficit for modeling multiple wakes and does not consider the superpositions of deflected wakes by upwind turbines. Since FLORIS benefits from the generic AD*

*model, it might be good to be evaluated with a LES model with the non-rotating AD model as well. Moreover, conducting such observations provides a clear view on the physics of the counter-rotating vortices.*

*44. P06. 08: The analysis was performed with a AD model of the turbine. The results look like the rotation of the rotor is included. Often this is not the case if an AD implementation is used. Therefore, this should be clearly stated when the model is introduced whether the rotation of the rotor is included or not. In case it is not included it should be commented on: To what extent could the results differ in a model that takes rotation into account? Is the use of "only" an AD model justified for this type of investigation.*

- Thank you also for these useful suggestions.  We have re-run the baseline, and yawed cases over again with rotation disabled (through a change to the SOWFA code).  This figure shows the wake "profiles" between the rotating and not rotating cases (taken 7D behind the downstream turbine) are very similar:

[Figure]

**Proposed solutions**

**8. How could the FLORIS model be potentially improved? Or which further analysis are required before such recommendations could be substantiated?**

- We have revised the conclusion to summarize what was shown in the paper, and then to discuss how improvements could be made and used:
  *In this paper, the role of flow structures, particularly a pair of counter-rotating vortices, generated in wind farm control through yaw misalignment was investigated. These vortices were shown to be important in (1) deforming the shape of the wake, (2) in explaining the asymmetry of efficacy in wake steering of oppositely signed yaw angles (rather than the currently used natural deflection angle), and (3) in understanding how a ``steered'' wake interacts with a downstream wake and with laterally adjacent wakes. These findings are critical for developing an effective, robust strategy for wind farm control.*

  *One important finding of the paper was that, because of the presence of these vortices, a steered wake can deflect the wake of downstream turbine, even if that turbine is not yawed, a process called ``secondary steering''. A second finding was that combinations of wakes from yawed turbines are shown to involve merging of generated cross flows. These combinations can lead to changes in power generation that are different then adding the changes of yawing individual turbines.*

  *In ongoing research at NREL, new engineering wake models which contain these behaviors due to the generated voritices are under development to be incorporated into FLORIS. Once completed these new models will be compared to similar SOWFA data sets to determine if they can resolve all discrepancies. Such new models, can then be used to develop new wind farm controllers. These new wind farm controllers can be optimized to take advantage of the controllability afforded by the vortices ability to manipulate the flow and will yield a more powerful form of wake steering than currently exists.*

**On the lack of wake center definition**

**5. P07. L03: "there does not appear to be any deflection" -> Due the colorbar it is**

**impossible to see any details of the flow structure. It indeed looks like no deflection of**

**the wake took place. But what is YOUR definition of wake? Could some wake tracking**

**be applied in order to make a reliable and clearer statements?**

*also*

*49. P09. L02: Mention the method used for obtaining the wake center*

- This sentence revised to:

- As a first qualitative observation of the baseline cases, there does not appear to be any natural deflection (this will be further examined later).
- Then below in the discussion of the power profiles, the analysis is expanded to discuss this using that figure for a more quantitative inspection of deflection.  We also add an additional paragraph to further emphasize the power profile comparison method used here, with comparison based on wake center identification:
- One advantage of using this method to compare wake predictions between models is that it focuses on the comparison on the quantity of interest, which is the power production of turbines at a given location.  Rather than trying to identify a wake center, focus is shifted from how far the centroid of deficit is shifted, to how much expected power production is possible at a given location.  This will be important, if for example, focus shifts from wake deflection to energy entrainment.

**Introduction and References to Literature**

*9. P01. L10: The first part should include a few more literature references. The*

*literature review on engineering models is limited just like the one about the interaction*

*of multiple wake.*

*Furthermore, there have been early wake deflection studies in the*

*wind tunnel which might be worth mentioning.*

- Thank you for this suggestion, this was a large oversight, and has been remedied, the following paragraph and references are added:
- *Early research in this field used wind tunnel experiments to demonstrate the possibility of wake steering.  Experiments presented in Dahlberg and Medici showed that for example, wake steering implemented in a two turbine row could yield a total relative power gain of 10%.  In Medici and Alfresson (2006), the wake of a model wind turbine in a tunnel was measured.  When the turbine was yawed, the wake was deflected, and additionally, vortex shedding was observed which was similar in behavior to what would be expected from solid discs.  In Wagenaar et al wake steering is studied at a scaled wind farm.*
- Dahlberg, J. and Medici, D.: Potential improvement of wind turbine array efficiency by active wake control, in: European Wind Energy Conference, 2003.
- Medici, D. and Alfredsson, P.: Measurements on a wind turbine wake: 3D effects and bluff body vortex shedding, Wind Energy, 9, 219–236, 2006.
- We have also discussion on the provided literature on new research in wake combination

*10. P01. L13: "Early work in wake steering . . ." This sentence somehow suggests that*

*first investigations of wake steering were performed using computational fluid dynamics.*

*In the years before, however, multiple investigations were carried out in the wind*

*tunnel. Please consider rephrasing this part.*

- This is rephrased, the above paragraph now defines the early work, and this paragraph now begins with "Later work…"

*11. P01. L22: "The model is shown to predict the behavior of wake steering and . . .". If*

*the model does predict the behavior of wake steering why do we need other models?*

*What are the weaknesses of the model? Consider adding further details or rephrase*

*the statement.*

- This sentence has been made more precise to say that the model accurately predicted wake steering for a given set of CFD simulations of particular conditions:
- *The model was shown to predict the behavior of wake steering for a given set of CFD simulations (focused largely on two-turbine, and fully waked scenarios) and, given its execution speed, can be used to design controllers as well as look at coupled wind farm layout and controls optimizations.*

*13. P03. L01: The 1st and 2nd objectives of the paper seem similar. They might be*

*combined or rephrased.*

- This paragraph has been re-written as follows, to track the change in emphasis through the title/abstract/discussion:
- *In this paper, a CFD-based analysis is used to examine how the consideration of the counter-rotating vortices can impact wind farm control analysis and design.  This paper undertakes an investigation of the impact of these vortices on yaw-based wake control.  The contributions of this paper are first a demonstration that a deflection-only control-oriented model of wind farm control can not reconcile all observed effects, even for to some extent for a single turbine wake case.  A second contribution is the demonstration that the influence of the vortices is especially critical when arrays of multiple turbines are considered.  A steered wake of an upstream turbine is shown to deflect the wake of an aligned turbine downstream, and combinations of steered turbines are shown to involve merging of generated cross flows.  The discussion section considers*

*how wind farm control, based on the generation of specific large-scale structures, and not on geometrical deflection, could be different and more effective than current methods. The future work recommendations conclude that incorporating the shed vortices into engineering models used to design wind farm controllers can improve performance and should be undertaken.*

*15. P03. L03 and L19: The 3rd objective is not addressed enough through the paper.*

*The described approach of a hypothetical wind turbine downstream (in section 3) is*

*introduced for reducing computational efforts of the simulation studies. It might be also*

*good to discuss the benefits of this method for wind farm control research.*

- The objectives paragraph has been reworded (also including review 2 comments) (along with abstract and conclusion.) Paragraph now reads:
  *In this paper, a CFD-based analysis is used to examine how the consideration of the counter-rotating vortices can impact wind farm control analysis and design. This paper undertakes an investigation of the impact of these vortices on yaw-based wake control. The contributions of this paper are first a demonstration that a deflection-only control-oriented model of wind farm control can not reconcile all observed effects, even for to some extent for a single turbine wake case. A second contribution is the demonstration that the influence of the vortices is especially critical when arrays of multiple turbines are considered. A steered wake of an upstream turbine is shown to deflect the wake of an aligned turbine downstream, and combinations of steered turbines are shown to involve merging of generated cross flows. The discussion section considers how wind farm control, based on the generation of specific large-scale structures, and not on geometrical deflection, could be different and more effective than current methods. The future work recommendations conclude that incorporating the shed vortices into engineering models used to design wind farm controllers can improve wind farm control performance and should be undertaken.*

*17. P03. L06: Who developed the two models? References*

- Development citations added for both FLORIS and SOWFA and explicit reference made to developers

*19. P03. L18: ".Churchfield et al." Reference Typo*

- This is fixed

*30. P04. L04: Reference for FAST*

- Reference added

**4. P05. L08: In the paper of Fleming 2015 this has only been investigated for the specific case that the upstream turbine is yawed and the downstream turbine has no lateral offset. Can you conclude that there are no cases where a negative yaw offset would be beneficial?**

- You are right, this sentence needs to be qualified in order to be accurate. It has been rephrased to:
- *For example, in the simulation study in Fleming 2015, where two 5MW turbines are aligned directly in the flow, after deducting the power lost because of yaw on the first turbine, only positive yaw produces an overall gain in power for a case of two turbines separated by 7 diameters (note this wouldn't be true for example in certain partial overlap cases).*

**21. P03. L20: In the 1st paragraph, some references for past research would be beneficial.**

- These examples have been added.

**35. P04. L27: Please distinguish between the approach of the current paper and the approach of the prior studies in the previous paragraph.**

- The following sentence has been added: *This way, rather than results from a handful of turbine positions, a continuum of turbine locations can be considered.*

**Technical questions**

**52. P09. L07: What about negatively yawed conditions?**

- We don't see a major discrepancy for this tuning of FLORIS, however, were we to re-tune FLORIS to remove the error on the positive side, the expectation is error would rise in the baseline and negative cases

**65. P15. L03: Is there a specific reason for choosing 3.5D?**

- This is halfway between all the rows. This could be for example 6D, the visualization is broadly similar.

*43. P06. 03: "cut-through slices of the flow at a distance downstream of the turbine."*

*-> flow = longitudinal velocity component?, vertical slice?*

- The following sentence is added: *These cut-through slices are cut cross-wise through the flow direction, and include the average value of all velocity components.*

*51. P09. L07: "Under positively . . ." Should it be "7D and more" instead of 5D? No*

*results are shown for distances greater than 5D and smaller than 7D.*

- This is corrected

*64. P14 L3: Please specify that Fig.13 corresponds to "YawAll" scenario. Moreover, a*

*bar graph helps to evaluate the results.*

- This is now specified.

*66. P16. L14: Are you not influencing the wake generation and due to this the characteristics*

*of the downstream flow. Therefore, the focus is still on controlling the wakes,*

*because they are your tool to influence the entire flow.*

- The conclusion has been reworded

**Figures**

**Horizontal Slices**

*25. Figure 1: Colorbar?*

- The colorbar is added

*26. Figure 1: Without a grid or ticks the use of the tick labels is limited*

- Tick marks have been added

27. *Figure 1: View from above down at the plants?*

- This is now specified in the caption

28. *Figure 1: Consider including a sketch to define positive and negative yaw offsets*

- The following text is added: Fig 1. illustrates a turbine in positive yaw in the convention used of positive being a counter-clockwise rotation when viewed from above.

33. *P04. L04: Please consider adding the coordinate system to Figure 1*

- Tick marks are added to clarify location, is this what is meant?

53. *Figure 6: Same comments as for Figure 1 (greyscale of colorbar, ticks, grid, . . .)*

- These matching changes have been made

54. *Figure 7: Same comments as for Figure 1 (greyscale of colorbar, ticks, grid, . . .),*

*add "at hub height" in the caption*

- These changes are made

**Cut-through slices**

37. *Figure 3: It should be clearly stated whether the wake flow is looked at from*

*upstream or downstream*

- Upstream, this is now indicated in the caption

38. *Figure 3: The color scheme should be avoided since information is lost when*

*printed in greyscale.*

- If it is acceptable to the reviewer, we would like to keep this color scheme.  The first reason is because there are places where the flow is slightly accelerated, and the red/blue format is helpful in seeing positive/negative changes.  The second reason is to be consistent with our other publications.
- To aid in visual clarity, several contour lines are overlayed, we hope this compensates

39. *Figure 3: Why are the limits of the colorbar -2 m/s to 2 m/s. Are there any positive*

*values at all?*

- This is chosen so that no change can be clearly indicated as gray. Also, because there are some positive increases to show (for example beneath the negatively steered wake).

*40. Figure 3: Grid lines or ticks should be added*

- These are added

*42. Figures 3,4,8,13: Please add a reference scale for the arrows to be able to see the magnitude of the vortices.*

- This scale is now included

*41. Figure 3 and Figure 4: Merging both figures would certainly make it easier for the reader to compare the impact of a positive and negative yaw misalignment on the shape of the wake.*

- Figures 3 and 4 are now merged

*45. Figure 4: Same comments as for Figure 3*

- Merged with 3

*55. Figure 8: Same comments as for Figure 3*

- Comments included for Fig 8 as well

*56. Figure 8: Caption "visualization of FLORIS 2D", add SOWFA*

- This is corrected

*57. Figure 8: YLabel: Everything is stated in "D". Here no explanation is given, the D is missing and X is introduced. Compare labeling with Figure 13*

- This has been fixed to match Fig 13, and visualization improvements from fig 3 and 4 included

**Hypothetical power plots**

*47. Figure 5: Caption: "showing good agreement . . ." -> Consider moving the analysis to the main text.*

- This was in text, so removed redundant text from caption

*48. Figure 5: yaxis labeling for right column missing*

- This is fixed

*58. Figure 9: Same comments as for Figure 5*

- This is fixed

**Grammar and Style and clarity**

*12. P02. L16: Typo: "to changing a the deficit"*

- Corrected

*16. P03. L06: "models of wind farms" -> wind farm models?*

- Corrected

*22. P03. L27: "( see Fig 1.)" Typo, unnecessary space before "see".*

- Corrected

*24. P03. L29: "by an additional hypothetical at some point" Typo, "turbine" missing?*

- Corrected

*32. P05. L03: "+/- 25âU¸e" The degree sign does not show up right in the reviewed ˚*

*draft. Please verify if this is a typo.*

- The appearance of the degree symbol has been corrected here and throughout the paper

*34. P05. L07: "For example, in Fleming . . ." Please consider rephrasing this sentence*

*due to its complex structure.*

- This sentence has been revised

*46. P07. L02: "Visual inspection indicates some discrepancies . . ." -> In the following*

*sentence an example about a similarity is given. Consider rephrasing.*

- This sentence is removed

*50. P09. L07: "Under positively . . ." Should it be positively offset?*

- This change has been made

*59. Figure 9: Caption: "Power of a hypothetical behind" -> Word missing "turbine"*

- This is corrected

*60. P13. L02: ". . . three-turbine cases, A . . ." Typo*

- This is corrected

*61. P13. L02: "A final case simulates a tightly spaced wind farm is used to further*

*explore . . ." -> Grammar?*

- This sentence has been adjusted

*63. P13 L17: The statement "we can compare the result of summing the effects of . . ."*

*is not clear enough. It might be reformulated better.*

- Revised to: *Then in the analysis, we can compare the results of summing the effects of individually yawed turbines from seperate simulations (``sumInd'') in post-processing to the case of simultaneously yawing all turbines within the simulation.*

**Response to review 2**

Thank you very much to the reviewer for the constructive and helpful criticism!

Responses the reviewer's points are included in red, and following is a version with track changes so actual changes can be easily identified.  Note some figures have been merged/added because of comments made by reviewer 1, and almost all figures have been updated.

**Descriptions of contributions and results**

1. The title is somehow misleading: It is clear after the reading why you chose the

title (your result #2, the suggestion), but actually don't really "control the flow" but

you analyze the effect of yawing wind turbines.

This is right, and we have made edits to the title, abstract, introduction to address this.  The paper has been retitled to "A simulation study demonstrating the importance of large-scale trailing vortices in wake steering".  And we hope all sections have been adjusted in a way which reflects this improved focus and clarity.

2. This brings me to another important point: you nicely describe new effects

which are very important for wind farm control. This deserves from my opinion

more credits. It is somehow included in your contribution #1 (demonstration that

FLORIS cannot describe it), but might be good to highlight it more in a positive

way.

Agreed, this point is now explicitly called out in the abstract and introduction.

3. In this way, your contributions #1 and #2 might be also be more specific to the

new effects: i) A upwind wake steers the wake of a un-yawed turbine (see Figure

6 and 2nd and 3rd row in Figure 11) and ii) and the wakes of wind turbines in

the same row have interactions which are unneglectable (sumInd different from

yawAll in Figure 11). The original contributions #1 and #2 are also overlapping.

4. The contributions #3 and #4 are very similar to your results #2 and #1. However, the finding and description of the new effects are missing in the listing of the results. Further, the points i) Suggestion: "we need to control the flow and not only steer the wakes" and ii) Motivation: "this requires new models" are important but still quite vague to be listed as the main results of a scientific paper.

These comments are also correct, and we have strived to improve the treatment of the contributions and conclusions with greater clarity. For example, the last paragraph of the introduction has been re-written as follows:

*In this paper, a CFD-based analysis is used to examine how the consideration of the counter-rotating vortices can impact wind farm control analysis and design. This paper undertakes an investigation of the impact of these vortices on yaw-based wake control. The contributions of this paper are first a demonstration that a deflection-only control-oriented model of wind farm control can not reconcile all observed effects, even for to some extent for a single turbine wake case. A second contribution is the demonstration that the influence of the vortices is especially critical when arrays of multiple turbines are considered. A steered wake of an upstream turbine is shown to deflect the wake of an aligned turbine downstream, and combinations of steered turbines are shown to involve merging of generated cross flows. The discussion section considers how wind farm control, based on the generation of specific large-scale structures, and not on geometrical deflection, could be different and more effective than current methods. The future work recommendations conclude that incorporating the shed vortices into engineering models used to design wind farm controllers can improve performance and should be undertaken.*

Which we hope more clearly underlines what is shown in the paper, and what is the subject of future work.

5. Addressing a similar issue: Your structure is: 1. Introduction, 2. Models, 3. Methods, 4-6 results, 7 conclusions. Sections 4-6 could be renamed to be more consistent, e.g. "one turbine case", "two turbine case", and "Multiple turbine case".

Or: "Results for two turbines", "results for three turbines", "results for a wind farm" or similar. The current section titles are a little bit confusing.

These suggestions are included and merged with suggestions from the reviewer 1 to clarify the section headings

**Missing details**

Since a large portion of the paper is to compare FLORIS to SOWFA and to show the

limits of it, it would be helpful to describe it in more details.

We agree, reviewer 1 made the same suggestion, and we have added details throughout the paper including:

- Additional details on FLORIS wake merging mechanism
- Details on SOWFA actuator disk model
- Details on atmospheric properties of the inflow used
- Additional references on FLORIS and SOWFA

**Other minor issues**

1. The ∘ sometimes is not a superscript, e.g. 2nd line of Section 4 and caption

Figure 3.

The degree signs have been regularized throughout the paper

2. Figure 3: Linking the arrow size to a wind speed might be helpful to get a better

feeling for the effects.

Yes, all images with cross-flow arrows now include a reference arrow of 1 m/s indicated

3. Figure 4: Period is missing at end of caption.

Figure 3 and 4 merged according to comment of review 1

4. Figure 5 and 9: ylabel of 2nd column is missing.

Added

And maybe the plots look

smoother when you use a higher discretization (now it seems to be 0.2 D).

They do, this is now done.

5. Figure 8: ylabel missing.

Added

And plots are too small. Maybe you could limit it to -2 D

and +7 D.

We've tried to improve the visibility of the figure, by including recommendations of reviewer 1 and additionally, adding contour lines, and reducing the density of quiver arrows.

6. Figure 9: "turbine" missing behind "hypothetical".

Fixed

7. Figure 10: The zero-indexing might be confusing. Why not turbine 1 to 12 and

"yaw 1" to "yaw 3"?

This is to be consistent with the turbine names within SOWFA, which is zero-indexed.  However, this can be changed of course in the text.

**A simulation study demonstrating the importance of large-scale trailing vortices in wake steering**

Paul Fleming[1], Jennifer Annoni[1], Matthew Churchfield[1], Luis Martinez[1], Kenny Gruchalla[1], Michael Lawson[1], and Patrick Moriarty[1]

[1]National Wind Technology Center, National Renewable Energy Laboratory, Golden, CO, 80401, USA

*Correspondence to:* Paul Fleming (paul.fleming@nrel.gov)

**Abstract.** In this paper, we investigate the role of flow structures generated in wind farm control through yaw misalignment. A pair of counter-rotating vortices are shown to be important in deforming the shape of the wake and in explaining the asymmetry of wake steering in oppositely signed yaw angles. We also demonstrate that vortices generated by an upstream turbine that is performing wake steering can deflect wakes of downstream turbines, even if they are themselves aligned.

We motivate the development of  improvements to control-oriented engineering models of wind farm control,  to include the effects of these large-scale flow structures. Such a new model would improve the predictability of control-oriented models.  Further, we demonstrate that the vortex structures created in wake steering can lead to greater impact on power generation than currently modeled in control-oriented models. We propose that wind farm controllers, can be made more effective if designed to take advantage of these effects.

**Copyright Statement**

The author's copyright for this publication is transferred to Alliance for Sustainable Energy, LLC. Alliance for Sustainable Energy, LLC is the manager and operator of the National Renewable Energy Laboratory. Employees of the Alliance for Sustainable Energy, LLC, under Contract No. DE-AC36-08GO28308 with the U.S. Dept. of Energy, have authored this work. The United States 
[revised manuscript text omitted]